# Category-Prompt Refined Feature Learning for Long-Tailed Multi-Label Image Classification

## ABSTRACT

Real-world data consistently exhibits a long-tailed distribution, often spanning multiple categories. This complexity underscores the challenge of content comprehension, particularly in scenarios requiring Long-Tailed Multi-Label image Classification (LTMLC). In such contexts, imbalanced data distribution and multi-object recognition pose significant hurdles. To address this issue, we propose a novel and effective approach for LTMLC, termed Category-Prompt Refined Feature Learning (CPRFL), utilizing semantic correlations between different categories and decoupling category-specific visual representations for each category. Specifically, CPRFL initializes category-prompts from the pretrained CLIP's embeddings and decouples category-specific visual representations through interaction with visual features, thereby facilitating the establishment of semantic correlations between the head and tail classes. To mitigate the visual-semantic domain bias, we design a progressive Dual-Path Back-Propagation mechanism to refine the prompts by progressively incorporating context-related visual information into prompts. Simultaneously, the refinement process facilitates the progressive purification of the category-specific visual representations under the guidance of the refined prompts. Furthermore, taking into account the negative-positive sample imbalance, we adopt the Asymmetric Loss as our optimization objective to suppress negative samples across all classes and potentially enhance the head-to-tail recognition performance. We validate the effectiveness of our method on two LTMLC benchmarks and extensive experiments demonstrate the superiority of our work over baselines.

## CCS CONCEPTS

• **Computing methodologies** → **Computer vision**; *Image representations*; • **Networks** → *Network architectures*.

## KEYWORDS

Multi-label classification, Long-tailed recognition, Visual-language pretrained models, Category-specific features, Interaction attention network, Prompt refined feature learning

## 1 INTRODUCTION

With the rapid development of deep networks, recent years have witnessed significant progress in computer vision, especially in

Permission to make digital or hard copies of all or part of this work for personal or classroom use is granted without fee provided that copies are not made or distributed for profit or commercial advantage and that copies bear this notice and the full citation on the first page. Copyrights for components of this work owned by others than the author(s) must be honored. Abstracting with credit is permitted. To copy otherwise, or republish, to post on servers or to redistribute to lists, requires prior specific permission and/or a fee. Request permissions from permissions@acm.org.

MM '24, October 28–November 1, 2024, Melbourne, Australia

© 2024 Copyright held by the owner/author(s). Publication rights licensed to ACM.
ACM ISBN 978-1-4503-XXXX-X/18/06
https://doi.org/XXXXXXX.XXXXXXX

image classification tasks [19, 26, 45, 47]. This progress greatly relies on many mainstream balanced benchmarks (e.g., CIFAR [25], ImageNet ILSVRC [12], MS COCO [31]), which have two key characteristics: 1) they provide a relatively balanced and sufficient number of samples across all classes, and 2) each sample belongs to only one category. However, in real-world applications, the distribution of different categories often follows a long-tailed pattern [34, 57], where deep networks tend to underperform on tail classes. Meanwhile, unlike the classical single-label classification, practical scenarios frequently involve images associated with multiple labels [9, 27, 32, 50], adding complexity and challenge to the task. To address these issues, an increasing number of works focus on the problem of Long-Tailed Multi-Label image Classification (LTMLC) [8, 16, 29, 52].

As the samples of tail classes are relatively scarce, mainstream methods for solving LTMLC focus on addressing the head-to-tail imbalance by employing various strategies, such as resampling the number of samples for each category [2, 3, 43, 52], re-weighting the loss for different categories [4, 11, 16, 51], and decoupling the learning of representation and classification head [23, 58]. For example, the Decoupling approach [23] designs four distinct sampling strategies evaluated for representation learning of long-tailed data. Similarly, Label-Distribution-Aware Margin (LDAM) [4] enforces class-dependent margin factors for different classes based on their training label frequencies. Although these methods have made significant contributions, they have often overlooked two crucial aspects. Firstly, it is of great importance to consider the semantic correlations between the head and tail classes in long-tailed learning. Leveraging such correlations can substantially improve the performance of tail classes with the support of head classes. Secondly, real-world images often encompass a variety of objects, scenes, or attributes, adding complexity to the classification task. The aforementioned methods typically consider extracting the visual representation of images from a global perspective. However, this global visual representation contains mixed features from multiple objects, hindering effective feature classification for each category. Therefore, how to explore semantic correlations between categories and extract local category-specific features in long-tailed data distributions remains a critical area of research.

Recently, Visual-Language Pretrained (VLP) models [40, 41, 59, 60] have been successfully adapted to various downstream visual tasks. For instance, CLIP [41], pretrained on billions of samples of image-text pairs, contains abundant linguistic knowledge from the Natural Language Processing (NLP) corpora in its text encoder. The text encoder demonstrates substantial potential in encoding semantic context representations within the text modality. Hence, it is feasible to leverage CLIP's text embedding representations to encode the semantic correlations between the head and tail classes. Furthermore, in numerous studies, CLIP's text embedding has been successfully employed as semantic prompts to decouple

local category-specific visual representations from global mixed features [7, 27, 32].

To tackle the challenges inherent in Long-Tailed Multi-Label Classification (LTMLC), we propose a novel and effective approach named **C**ategory-**P**rompt **R**efined **F**eature **L**earning (CPRFL). CPRFL leverages CLIP's text encoder to extract category semantics, thereby enabling the establishment of semantic correlations between the head and tail classes. This is achieved through the robust semantic representation capabilities of CLIP's text encoder. Subsequently, the extracted category semantics are utilized to initialize prompts for all categories, which interact with visual features in order to discern context-related visual information specific to each category. Such visual-semantic interaction can effectively decouple category-specific visual representations from the input samples. However, these initial prompts lack visual-context information, resulting in a significant data bias between the semantic and visual domains during information interaction. In essence, the initial prompts may not be precise, thereby compromising the quality of category-specific visual representations. To mitigate this issue, we introduce a progressive Dual-Path Back-Propagation mechanism to iteratively refine the prompts. This mechanism progressively accumulates context-related visual information into the prompts. Concurrently, the category-specific visual representations are purified under the guidance of the refined prompts, enhancing their relevance and accuracy. Finally, to further address the negative-positive imbalance problem inherent in multiple categories, we incorporate the Re-Weighting (RW) strategy, commonly utilized in such scenarios. Specifically, we employ the Asymmetric Loss (ASL) [42] as our optimization objective, which effectively suppresses negative samples across all classes and potentially improves head-to-tail category performance in LTMLC tasks.

The contributions of our work can be summarized as follows:

- We propose a novel prompt-learning approach termed Category Prompt Refined Feature Learning (CPRFL), for Long-Tailed Multi-Label image Classification (LTMLC). CPRFL leverages CLIP's text encoder to extract category semantics, harnessing its powerful semantic representation capability. This facilitates the establishment of semantic correlations between the head and tail classes. The extracted category semantics serve as category-prompts to enable the decoupling of category-specific visual representations. To the best of our knowledge, this is the first work to utilize category semantic correlations to mitigate the head-to-tail imbalance problem in LTMLC, offering a pioneering solution tailored to the distinctive characteristics of the data.
- We design a progressive Dual-Path Back-Propagation mechanism aimed at refining the category-prompts by progressively incorporating context-related visual information into prompts during visual-semantic interaction. By employing a series of dual-path gradient back-propagations, we effectively counteract the visual-semantic domain bias stemming from the initial prompts. Simultaneously, the refinement process facilitates the progressive purification of category-specific visual representations.

- We conduct experiments on two LTMLC benchmarks, including the publicly available datasets COCO-LT and VOC-LT . Extensive experiments not only validate the effectiveness of our approach but also highlight its significant superiority over the recent state-of-the-art approaches.

## 2 RELATED WORK

### 2.1 Long-Tailed Visual Recognition

Real-world data often follows a long-tailed distribution, presenting a significant challenge for traditional models due to the imbalanced class distributions. Common methods to address this challenge include direct resampling of training samples to balance category distribution, which may result in over-fitting of the tail categories [2, 3, 18, 43]. Another strategy involves loss re-weighting based on label frequencies of training samples to rebalance the uneven positive gradients among classes [4, 11, 21, 51]. More recently, researchers have explored some techniques like transfer learning [34, 61] and self-supervised learning [22, 56] to address imbalanced class distribution. As the Vision-Language Pretrained (VLP) models like CLIP [41] exhibit strong zero-shot adaptation performance, some strategies have been proposed to adapt them to downstream long-tailed learning tasks [13, 35, 44, 48]. These VLP-based methods incorporate additional language data to generate auxiliary confidence scores and fine-tune the CLIP-based model on long-tailed data. However, the enhanced performance of tail classes in these methods is largely attributed to CLIP's image encoder, which has been pretrained on a vast dataset containing many tail samples. This reliance on the image encoder may inadvertently leverage prior exposure to a large amount of visual data. In our work, we specifically utilize only CLIP's text encoder to extract category semantics, allowing us to avoid this potential bias and establish a more robust method for addressing the long-tailed problem.

### 2.2 Multi-Label Image Classification

For the Multi-Label image Classification (MLC) tasks, early solutions involved training separate binary classifiers for each label [49]. Consequently, recent research addresses the MLC problem by utilizing category semantics to model label semantic correlations. CNN-based methods [6, 15, 50, 60] utilize Recurrent Neural Networks (RNNs) to extract features sensitive to label dependencies, implicitly capturing these dependencies. Similarly, statistical co-occurrence graphs are constructed to leverage Graph Convolutional Network (GCNs) for representing label correlations [7, 9, 36, 54]. For example, SSGRL [7] incorporates category semantics to guide learning semantic-specific representations and explores their interactions via a graph propagation mechanism. With the rise of Transformer across various computer vision tasks, Transformer-based methods [27, 32, 38] utilize the core attention mechanisms in Transformer [33] to explore label correlations. For instance, C-Tran [27] leverages category semantics extracted from word embedding to facilitate visual-semantic interaction within the Transformer. However, these methods often have complex designs and may not generalize well to LTMLC. Recently, VLP models [41] have been adapted for downstream MLC tasks to address few-shot and zero-shot problems [1, 17, 20]. Neglecting the imbalanced distribution of samples, they either have poor generalization on the LTMLC tasks.

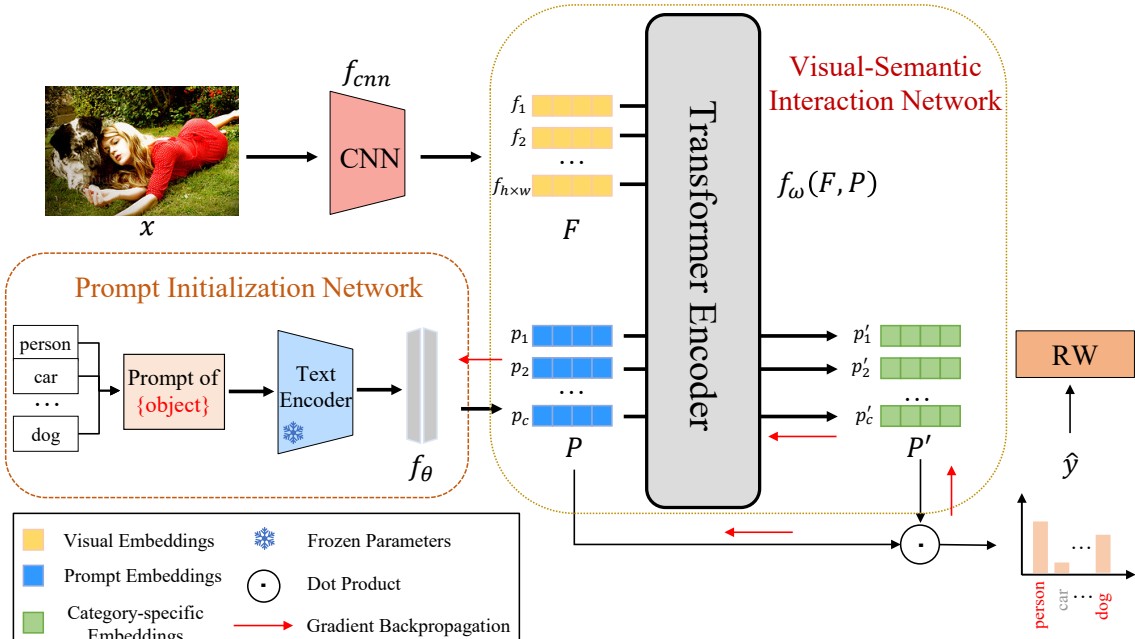

**Figure 1: Overall framework of our CPRFL for long-tailed multi-label image classification. Overall, our approach consists of two sub-networks: *Prompt Initialization (PI) network* and *Visual-Semantic Interaction (VSI) network*. The initial prompts $P$ are extracted from CLIP's text embedding within the PI network, and then these prompts are employed to interact with visual features $F$ within the VSI network, facilitating the decoupling of category-specific visual representations $P'$. Finally, we compute the similarities between category-specific features $P'$ and corresponding prompts $P$ to obtain the prediction probability for each category and utilize a progressive Dual-Path Back-Propagation mechanism to refine the prompts. To further address the negative-positive imbalance problem inherent in multiple categories, we incorporate a Re-Weighting (RW) strategy.**

## 2.3 Long-Tailed Multi-Label Image Classification

In recent years, research on addressing class imbalance in multi-label settings has been relatively limited. Similar to re-weighting strategies, Wu *et al.* [52] propose a distribution-balanced (DB) loss to slow down the optimization rate of negative labels based on binary-cross-entropy loss. Based on DB loss, Lin *et al.* [29] aim to flexibly adjust the training probability and further reduce the probability gap between positive and negative labels. To adapt re-sampling strategies to multi-label settings, Chen *et al.* [8] introduce a group sampling strategy, while Guo *et al.* [16] adopt collaborative training on both uniform and re-balanced samplings to alleviate the class imbalance. Additionally, a prompt-tuning method [53] has been proposed to adapt pretrained CLIP [41] to LTMLC. However, these methods may partly compromise the performance of the head classes while improving tail classes. In this paper, we utilize pretrained CLIP's text embedding to extract category semantics and guide visual-semantic information interaction, which explicitly establishes semantic correlations between the head and tail classes and decouple category-specific visual representations,leading to synchronous improvements in head-to-tail performance in LTMLC.

## 3 METHODS

### 3.1 Overview

In this section, we introduce the proposed CPRFL approach, consisting of two sub-networks, i.e., *Prompt Initialize (PI) network* and *Visual-Semantic Interaction (VSI) network*. Firstly, we leverage the pretrained CLIP's text embedding to initialize category-prompts within the PI network, utilizing category semantics to encode semantic correlations between different categories. Subsequently, these initialized prompts interact with extracted visual features using a Transformer encoder within the VSI network. This interaction process facilitates the decoupling of category-specific visual representations, enabling the framework to discern context-related visual information associated with each category. Finally, we compute the similarity between the category-specific features and their corresponding prompts at the category level to obtain the prediction probability for each category. To mitigate the visual-semantic domain bias, we employ a progressive Dual-Path Back-Propagation mechanism guided by category-prompt learning to refine the prompts and progressively purify the category-specific visual representations over the training iterations. To further address negative-positive imbalance issue, we optimize the framework adopting a Re-Weighting strategy (i.e., an Asymmetric loss (ASL) [42]), which helps suppress negative samples across all classes. Fig. 1 illustrates the entire pipeline of the proposed CPRFL approach.

**Feature Extraction.** Given an input image $x$ from the dataset $D$, we first utilize a backbone network to extract local image features $f_{loc}^x \in \mathbb{R}^{h \times w \times d_0}$, where $d_0, h, w$ denote the number of channels, height and width, respectively. In this paper, we employ a convolutional network such as ResNet-101 [19] and obtain the local features by removing the last pooling layer. After that, we add a linear layer $\varphi$ to project the features from dimension $d_0$ to $d$ into a visual-semantic joint space to match the dimension of category-prompts:

$$\mathcal{F} = \varphi(f_{loc}^x) = \{f_1, f_2, ..., f_v\} \in \mathbb{R}^{v \times d}, v = h \times w. \tag{1}$$

Utilizing the local features, we conduct visual-semantic information interaction between them and the initial category-prompts to discern category-specific visual information.

**Semantic Extraction.** Formally, the pretrained CLIP [41] comprises an image encoder $f(\bullet)$ and a text encoder $g(\bullet)$. For our purpose, we solely utilize the text encoder to extract category semantics. Specifically, we adopt a classic predefined template "a photo of a [CLASS]" as the input text for the text encoder. Then the text encoder maps the input text (class $i$, $i = 1, ..., c$) to the text embedding $\mathcal{W} = g(i) = \{w_1, w_2, ..., w_c\} \in \mathbb{R}^{c \times m}$, where $c$ represents the number of classes, and $m$ denotes the dimension length of the embedding. This extracted text embedding serves as the category semantics for initializing the category-prompts.

## 3.2 Category-Prompt Initialization

In order to bridge the gap between the semantic and visual domains, recent work [27, 50] has attempted to project semantic word embeddings into a visual-semantic joint space using linear layers. Instead of emloying linear layers for this projection directly, we opt for nonlinear structures to handle category semantics derived from pretrained CLIP's text embedding. This approach allows us to achieve a more sophisticated projection from the semantic space to the visual-semantic joint space.

Specifically, we design a Prompt Initialize (PI) network, which consists of two fully connected layers followed by a nonlinear activation function. Through the nonlinear transformation performed by the PI network, we map the pretrained CLIP's text embedding $\mathcal{W}$ to the initial category-prompts $\mathcal{P} = \{p_1, p_2, ..., p_c\} \in \mathbb{R}^{c \times d}$. Eq. 2 illustrates the entire initialization process:

$$\mathcal{P} = GELU(\mathcal{W}W_1 + b_1)W_2 + b_2, \tag{2}$$

where $W_1, W_2, b_1, b_2$ denote the weight matrices and bias vectors of the two linear layers, and $GELU$ represents the nonlinear activation function. Here, $W_1 \in \mathbb{R}^{m \times t}$, $W_2 \in \mathbb{R}^{t \times d}$, $t = \tau \times d$ and $\tau$ is the expansion coefficient controlling the dimension of hidden layers. Typically, $\tau$ is set to 0.5 in our experiments, and we will investigate its impact further in the supplementary material.

The PI network plays a crucial role in extracting category semantics from the pretrained CLIP's text encoder, leveraging its powerful semantic representation capability to establish semantic correlations between different categories without relying on ground-truth labels. By initializing category-prompts with category semantics, the PI network facilitates the projection from the semantic space to the visual-semantic joint space. Additionally, the nonlinear design of our PI network enhances the visual-semantic interaction

capacity of the extracted category-prompts, thereby improving the subsequent visual-semantic information interaction.

## 3.3 Visual-Semantic Information Interaction

With the widespread adoption of Transformer in the field of computer vision, as evidenced by recent works [10, 27, 32] showcasing the capability of typical attention mechanisms to enhance the interaction between visual-semantic cross-modal features, we are motivated to design a Visual-Semantic Interaction (VSI) network. This network incorporates a Transformer encoder, which takes as input the initial category-prompts and visual features. The Transformer encoder performs visual-semantic information interaction to discern context-related visual information specific to each category. This interaction process effectively decouples category-specific visual representations, thereby facilitating better feature classification for each category.

To facilitate visual-semantic information interaction between category-prompts and visual features, we concatenate the initial category-prompts $\mathcal{P} \in \mathbb{R}^{c \times d}$ with the visual features $\mathcal{F} \in \mathbb{R}^{v \times d}$, forming a combined set of embeddings $Z = (\mathcal{F}, \mathcal{P}) \in \mathbb{R}^{(v+c) \times d}$. These embeddings are then input to the VSI network for the visual-semantic information interaction. Within the VSI network, each embedding $z_i \in Z$ undergoes calculation and updating through the multi-head self-attention mechanism inherent to the Transformer encoder. Notably, we focus solely on updating the category-prompts $\mathcal{P}$, as these represent the decoupling part of category-specific visual representations. The attention weight $\alpha_{ij}^p$ and the subsequent updating process are computed as follows:

$$\alpha_{ij}^p = softmax\left((W_q p_i)^T (W_k z_i)/\sqrt{d}\right), \tag{3}$$

$$\bar{p}_i = \sum_{j=1}^{} (\alpha_{ij}^p W_v z_j), \tag{4}$$

$$p_i' = GELU(\bar{p}_i W_r + b_3)W_o + b_4, \tag{5}$$

where $W_q, W_k, W_v$ are the query, key, and value weight matrices, $W_r, W_o$ are transformation matrices, and $b_3, b_4$ are bias vectors. To streamline the complexity of the VSI network, we opt for one single layer of Transformer encoders without stacking. The resulting output of the VSI network and the category-specific visual features are denoted as $Z' = \{f_1', f_2', ..., f_v', p_1', p_2', ..., p_c'\}$ and $\mathcal{P}' = \{p_1', p_2', ..., p_c'\}$, respectively. Within the self-attention mechanism described in Eq. 3, each category-prompt embedding comprehensively considers its attention towards all local visual features and other category-prompt embeddings. This comprehensive attention mechanism effectively discerns context-related visual information from the samples, leading to the decoupling of category-specific visual representations.

## 3.4 Category-Prompt Refined Feature Learning

Following the interaction between visual features and initial prompts via the VSI network, the resulting output $\mathcal{P}'$ serves as category-specific features for classification. In traditional Transformer-based methods [10, 27, 32], the specific output features obtained from the Transformer are typically projected onto the label space using linear layers for final classification. Different from these methods, we employ the category-prompts $\mathcal{P}$ as the classifier and compute the

(a) Dual-Path Gradient Back-Propagation          (b) Category-Prompt Refined Feature Learning

**Figure 2: The refined learning process for category-prompts and category-specific features. We employ a progressive Dual-Path Back-Propagation mechanism to refine the prompts and progressively purify the category-specific visual representations over the training iterations. The depth of color represents the accuracy of the features, and the darker the color, the higher the accuracy.**

similarity between the category-specific features and the category-prompts to conduct classification within the feature space. The classification probability $s_i$ for class $i$ can be calculated by Eq. 6:

$$s_i = sigmoid(p'_i \cdot p_i). \qquad (6)$$

In the context of the unique data characteristics intrinsic to the multi-label setting, we compute the dot-product similarity between the category-specific feature vector of each category and the corresponding prompt vector to determine the probability, which computes absolute similarity. We deviate from the traditional similarity pattern using relative measurement between category-specific feature vector of each category and all prompt vectors. The reason is to mitigate the computational redundancy incurred by calculating similarity between the feature vector of each category and unrelated category prompts, which is unnecessary.

The initial prompts lack crucial visual-context information, leading to a substantial data bias between the semantic and visual domains during information interaction. This discrepancy results in imprecise initial prompts, consequently affecting the quality of category-specific visual representations. To mitigate this issue, we introduce a progressive Dual-Path Back-Propagation mechanism guided by category-prompt learning. This mechanism involves two paths of gradient optimization during model training (shown in Figure 2a): one path through the VSI network and another path directly to the PI network. The former path also optimizes the VSI network to enhance its ability for visual-semantic information interaction. By employing a series of dual-path gradient back-propagations, the prompts are gradually refined over the training iterations, allowing for the progressive accumulation of context-related visual information. Concurrently, the refined prompts guide the generation of more accurate category-specific visual representations, leading to the progressive purification of category-specific features. We term this entire process "Prompt Refined Feature Learning", and it will be iteratively conducted until convergence, as illustrated in Figure 2b.

## 3.5 Optimization

To further address the negative-positive sample imbalance inherent in multiple categories, we integrate the Re-Weighting (RW) strategy, commonly utilized in such scenarios. Specifically, we adopt the Asymmetric Loss (ASL) [42] as our optimization objective, which is a variant of focal loss [30] with different $\gamma$ values for positive and negative samples. Given an input image $x_i$, our model predicts its final category probabilities $S_i = \{s^i_1, s^i_2, ..., s^i_c\}$ and its ground truth is $Y_i = \{y^i_1, y^i_2, ..., y^i_c\}$. We train the whole framework using ASL as shown in Eq. 7:

$$\mathcal{L}_{cls} = \mathcal{L}_{ASL} = \sum_{x_i \in X} \sum_{j=1}^{c} \begin{cases} (1 - s^i_j)^{\gamma^+} log(s^i_j), & s^i_j = 1, \\ (\tilde{s}^i_j)^{\gamma^-} log(1 - \tilde{s}^i_j), & s^i_j = 0, \end{cases} \qquad (7)$$

where $c$ is the number of classes. $\tilde{s}^i_j$ is the hard threshold in ASL, denoted as $\tilde{s}^i_j = max(s^i_j - \mu, 0)$. $\mu$ is a threshold used to filter out negative samples with low confidence. By default, we set $\gamma^+ = 0$ and $\gamma^- = 4$ in our experiments. In our framework, ASL effectively suppresses negative samples across all classes, potentially improving head-to-tail category performance in LTMLC tasks.

## 4 EXPERIMENTS

### 4.1 Experimental Setup

**Datasets.** Following the settings outlined in [52], we conduct experiments on two datasets for long-tailed multi-label visual recognition: VOC-LT and COCO-LT. These two datasets are artificially sampled from two well-known multi-label recognition benchmarks: Pascal VOC [14] and MS-COCO [31], respectively.

**VOC-LT** is created from the 2012 train-val set of Pascal VOC, following the guidelines provided in [52]. The training set comprises 1,142 images annotated with 20 class labels. The number of images per class varies from 4 to 775. To simulate a long-tailed distribution, all classes are categorized into three groups based on the number of training samples per class: head classes (more than 100 samples), medium classes (20 to 100 samples), and tail classes (less than 20 samples). After splitting, the ratio of head, medium, and tail classes

**Table 1: The mAP (%) performance of the proposed CPRFL and comparison methods on two long-tailed multi-label datasets. We present the mAP results on overall, head, medium, and tail classes. "CPRFL-GloVe" denotes our CPR with GloVe word embedding, and "CPR-CLIP" denotes our CPRFL with CLIP-RN50 's text embedding. Bold indicates the best scores.**

| Datasets | | VOC-LT | | | | COCO-LT | | | |
|---|---|---|---|---|---|---|---|---|---|
| Methods | | total | head | medium | tail | | total | head | medium | tail |
| ERM | | 70.86 | 68.91 | 80.20 | 65.31 | | 41.27 | 48.48 | 49.06 | 24.25 |
| RW | | 74.70 | 67.58 | 82.81 | 73.96 | | 42.27 | 48.62 | 45.80 | 32.02 |
| ML-GCN [9] | | 68.92 | 70.14 | 76.41 | 62.39 | | 44.24 | 44.04 | 48.36 | 38.96 |
| OLTR [34] | | 71.02 | 70.31 | 79.80 | 64.95 | | 45.83 | 47.45 | 50.63 | 38.05 |
| LDAM [4] | | 70.73 | 68.73 | 80.38 | 69.09 | | 40.53 | 48.77 | 48.38 | 22.92 |
| CB Focal [11] | | 75.24 | 70.30 | 83.53 | 72.74 | | 49.06 | 47.91 | 53.01 | 44.85 |
| BBN [58] | | 73.37 | 71.31 | 81.76 | 68.62 | | 50.00 | 49.79 | 53.99 | 44.91 |
| DB Focal [52] | | 78.94 | 73.22 | 84.18 | 79.30 | | 53.55 | 51.13 | 57.05 | 51.06 |
| ASL [42] | | 76.40 | 70.70 | 82.26 | 76.29 | | 50.21 | 49.05 | 53.65 | 46.68 |
| LTML [16] | | 81.44 | 75.68 | 85.53 | 82.69 | | 56.90 | 54.13 | 60.59 | 54.47 |
| CDRS+AFL [46] | | 78.96 | 73.35 | 85.03 | 78.63 | | 55.35 | 52.45 | 59.48 | 52.46 |
| Bilateral-TPS [28] | | 81.58 | 75.88 | 84.11 | 83.95 | | 56.38 | 55.93 | 58.26 | 54.29 |
| PG Loss [29] | | 80.37 | 73.67 | 83.83 | 82.88 | | 54.43 | 51.23 | 57.42 | 53.40 |
| COMIC [55] | | 81.53 | 73.10 | 89.18 | 84.53 | | 55.08 | 49.21 | 60.08 | 55.36 |
| CAE-Net [5] | | 81.61 | 74.00 | 85.35 | 85.28 | | 57.64 | 52.37 | 61.18 | 57.63 |
| CPRFL-GloVe(ours) | | 85.14 | **82.50** | 90.42 | 83.17 | | 65.18 | 65.12 | 69.97 | 58.91 |
| CPRFL-CLIP(ours) | | **86.28** | 81.84 | **90.51** | **86.43** | | **66.69** | **66.35** | **70.99** | **61.33** |

becomes 6:6:8. To evaluate the model's performance, we employ the VOC 2007 test set, which consists of 4,952 images.

Similarly, **COCO-LT** is derived from the MS-COCO 2017 dataset using a comparable approach. The training set of COCO-LT comprises 1,909 images annotated with 80 class labels. The number of images per class ranges from 6 to 1,128. The distribution of head, medium, and tail classes in COCO-LT is set to 22:33:25. The performance evaluation is conducted on the MS-COCO 2017 test set, which contains 5,000 images.

***Implementation Details.*** For the pretrained CLIP models, we adopt CLIP ResNet-50 or ViT-Base/16 [41] and use the corresponding CLIP Transformer as the text encoder. For network optimization, we use the Adam optimizer [24] with a weight decay of $1e − 4$ and the training epochs are set to 30. The batch size is 32, and the learning rates for COCO-LT, and VOC-LT are empirically initialized with $1e − 5$, $5e − 5$. We use mean average precision (**mAP**) as the evaluation metric to assess the performance of long-tailed multi-label visual recognition across all classes. More implementation details can be found in the supplementary material.

## 4.2 Experimental Results

To validate the effectiveness of our proposed method, we compare it with previous state-of-the-art methods on two long-tailed multi-label datasets. The state-of-the-art methods include popular long-tailed learning methods and multi-label algorithms, i.e., Empirical Risk Minimization (ERM), a smooth version of Re-Weighting (RW) using the inverse proportion to the square root of class frequency, ML-GCN [9], OLTR [34], LDAM [4], Class-Balanced (CB) Focal [11], BBN [58], Distribution-Balanced (DB) Focal [52], ASL [42]. We also compare our CPRFL with previous LTMLC methods, including:

- LTML [16]: collaborative training on the uniform and rebalanced samplings with learnable logit compensation.
- CDRS+AFL [46]: the combination of copy-decoupling resampling strategy and adaptive weighted focal loss.
- Bilateral-TPS [28]: the modification of a bilateral structure that samples both the original and proposed sampling distributions to represent tail classes.
- Probability Guided (PG) Loss [29]: an improved version of DB focal that can flexibly adjust the training probability and further reduce the probability gap between positive and negative labels.
- COMIC [55]: an end-to-end learning framework that corrects missing labels, adaptively modifies the attention to different samples, and balances the classifier on head-to-tail learning.
- CAE-Net [5]: a class-aware embedding network that learns robust class-based representations.

For the proposed CPRFL, we present the results using different text embeddings for fair comparisons, i.e., GloVe word embedding [39] ("CPRFL-Glove") and CLIP's text embedding ("CPRFL-CLIP"). Notably, we don't compare our CPRFL with previous CLIP-based methods, because the superior performances of tail classes in these methods may be due to the image encoder pretrained on large-scale datasets, which may potentially involve a considerable number of tail-category samples and thereby lead to an unfair comparison. Table 1 illustrates the mAP performance of different methods. Experimental results show that our proposed method outperforms previous methods by a large margin. Specifically, our proposed CPRFL achieves outstanding results, with total mAP scores of 86.28% and 66.69% on VOC-LT and COCO-LT, respectively. Compared to the previous SOTA method (CAE-Net), CPRFL exhibits notable improvements of approximately 4.67% and 9.05% in total mAP on

**Table 2: The mAP (%) performance of the proposed CPRFL with different multi-label classification losses on two long-tailed multi-label datasets. Bold indicates the best scores.**

| Dataset | VOC-LT | | | |
|---|---|---|---|---|
| Loss Functions | total | head | medium | tail |
| BCE | 79.60 | 82.37 | 88.95 | 70.52 |
| MLS | 80.75 | 82.44 | 90.37 | 72.26 |
| CB Focal [11] | 83.87 | 80.32 | 90.48 | 81.54 |
| DB No-Focal [52] | 84.27 | **82.64** | 89.06 | 81.89 |
| DB Focal [52] | 86.18 | 81.81 | 90.01 | 86.30 |
| ASL [42] | **86.28** | 81.84 | **90.51** | **86.43** |

| Dataset | COCO-LT | | | |
|---|---|---|---|---|
| Loss Functions | total | head | medium | tail |
| BCE | 59.62 | 61.48 | 66.09 | 49.47 |
| MLS | 59.91 | 63.36 | 66.19 | 48.59 |
| CB Focal [11] | 64.20 | 64.50 | 69.36 | 57.12 |
| DB No-Focal [52] | 64.28 | 64.31 | 69.62 | 57.21 |
| DB Focal [52] | 65.12 | 63.74 | 69.97 | 59.91 |
| ASL [42] | **66.69** | **66.35** | **70.99** | **61.33** |

VOC-LT and COCO-LT. Moreover, CPRFL excels in all three class subsets (head, medium, and tail classes), with particularly noteworthy gains observed in tail classes, achieving a remarkable 3.7% mAP increase on COCO-LT. In VOC-LT, performance enhancements are primarily observed in both head and medium classes. These results fully demonstrate the efficacy of CPRFL in achieving synchronous improvements in head-to-tail recognition performance.

## 4.3 Ablation Study

To thoroughly investigate the impact of each component within the proposed CPRFL framework, we conducted a series of ablation studies on the influence of classification loss, category semantics used for prompt initialization, and the components of CPRFL.

**Multi-Label Classification Loss**. To further address the negative-positive sample imbalance problem, we adopt the Asymmetric Loss (ASL) [41] as the multi-label classification loss, which effectively suppresses negative samples across all classes. In this part, we compare several classification losses for optimizing our CPRFL approach, including Binary Cross-Entropy Loss (BCE), Multi-Label Soft Margin Loss (MSL), Class-Balanced Focal Loss (CB Focal) [11], Distribution-Balanced Loss (DB Focal) [52], a No-Focal version of DB Loss, and ASL [41]. Table 2 lists the comparison results on VOC-LT and COCO-LT. The results demonstrate that ASL consistently outperforms other classification losses for the LTMLC tasks. This superior performance can be attributed to ASL's ability to account for the dominance of negative-positive imbalance across all classes in LTMLC, a key aspect that the other losses do not adequately address.

**Category Semantics for Prompt Initialization**. We further conduct a comparative analysis of CPRFL's performance with various types of category semantics for prompt initialization to explore whether the notable improvement of our approach is primarily attributed to the powerful pretrained linguistic knowledge in

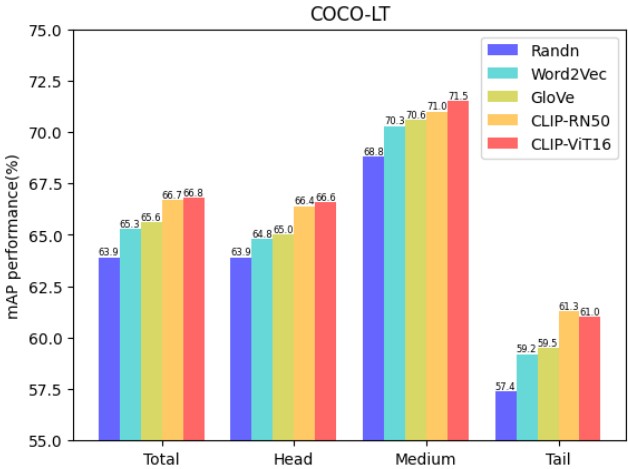

**Figure 3: The mAP (%) performance with various types of category semantics for prompt initialization on COCO-LT dataset.**

CLIP. Specifically, we evaluate three different word embeddings, i.e., Randn (random initialization embedding), Word2Vec [37], GloVe [39], and CLIP-RN50, CLIP-ViT16. The results on COCO-LT are presented in Figure 2. From the figure, it is clear that CPRFL utilizing CLIP outperforms the Word2Vec and GloVe embeddings across all classes. This can be attributed to CLIP's multi-modal pre-training, which enhances the alignment of semantic embedding with visual concepts and strengthens the representation ability of category semantics. Additionally, our CPRFL with Randn embedding (63.9%) surpasses all previous state-of-the-art methods (57.6%) as reported in Table 1. This suggests that even without leveraging CLIP's pre-trained semantic knowledge, our CPRFL approach still outperforms other methods in the LTMLC tasks.

**Components Analysis of CPRFL**. To evaluate the contributions of various components to our method for long-tailed multi-label classification, we conduct a series of ablation studies, with the results summarized in Table 3. Our baseline experiment uses focal loss [30] and achieves mAP performances of 73.88% on VOC-LT and 49.46% on COCO-LT. Our CPRFL framework comprises two key components: Visual-Semantic Interaction (VSI) and Prompt Initialization (PI). VSI leverages category-semantic prompts to decouple category-specific visual representations from samples, while PI establishes semantic correlations between different categories using CLIP. Adding these two components leads to significant improvements across head, medium, and tail classes, and the overall mAP outperforms the baseline by 10.34% on VOC-LT and 14.80% on COCO-LT. We attribute these gains to the Dual-Path Back-Propagation mechanism guided by this two components, which can refine the category-prompts and enhance visual-semantic interaction, thereby progressively purifying category-specific features during iterative training. Additionally, integrating VSI, PI with the Re-Weighting (RW) strategy further improves mAP performance, particularly for the tail classes, with increases of 4.27% on VOC-LT and 4.13% on COCO-LT.

Table 3: The ablation analysis on different components of the proposed CPRFL. Here "VSI" denotes Visual-Semantic Interaction, "PI" denotes Prompt Initialization, "RW" denotes Re-Weighting strategy, "avg.△" denotes average performance improvement. Bold indicates the best scores.

| | VSI | PI | RW | VOC-LT | | | | avg.△ | COCO-LT | | | | avg.△ |
|---|---|---|---|---|---|---|---|---|---|---|---|---|---|
| | | | | total | head | medium | tail | | total | head | medium | tail | |
| Baseline | | | | 73.88 | 69.41 | 81.43 | 71.56 | | 49.46 | 49.80 | 54.77 | 42.14 | |
| | √ | | | 83.87 | 80.32 | 90.53 | 81.54 | +9.99 | 63.87 | 63.55 | 69.01 | 57.36 | +14.40 |
| | √ | √ | | 84.24 | 80.78 | 90.47 | 82.16 | +10.34 | 64.27 | 64.30 | 69.62 | 57.20 | +14.80 |
| | √ | √ | √ | 86.28 | 81.84 | 90.51 | 86.43 | **+12.20** | 66.69 | 66.35 | 70.99 | 61.33 | **+17.30** |

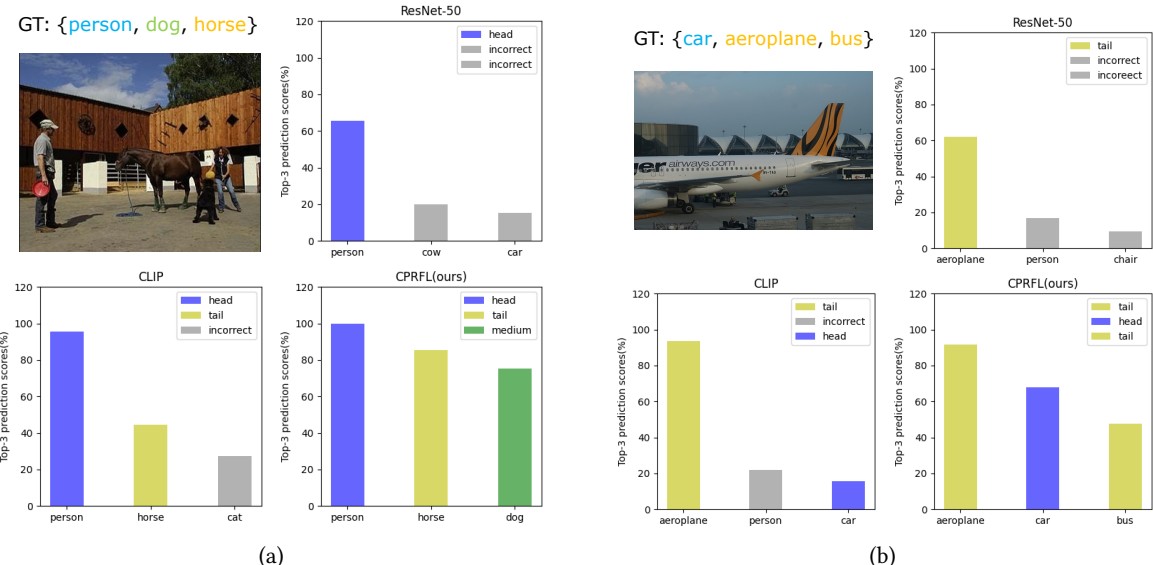

(a)                     (b)

Figure 4: Visualization examples of Top-3 predicated categories by ResNet-50, CLIP and our CPRFL.

## 4.4 Qualitative Analysis

To better understand how our method handles long-tailed multi-label data, we conduct qualitative experiments using ResNet-50, CLIP, and our proposed CPRFL. Figure 4 visualizes the top-3 category predictions from these different models, with CPRFL demonstrating superior performance, particularly for the tail classes. In Figure 4a, ResNet-50 solely recognizes the head class [person] but fails to classify the tail class [horse], which is a common challenge faced by visual models lacking semantic guidance. The emergence of CLIP is a great remedy for this issue, owing to its strong semantic linguistic supervision. However, CLIP relies on global visual representations from the image encoder and lacks the ability to decouple category-specific features, which may lead to overlooking small objects. In contrast, our CPRFL not only provides more accurate category predictions but also achieves higher prediction scores for the tail class [horse]. This is due to CPRFL's ability to leverage semantic correlations between the head and tail classes, like [person] and [horse], to boost the prediction probability for [horse]. Additionally, by effectively decoupling category-specific features, our method can recognize small objects such as [dog], a medium class. As a result, our proposed CPRFL demonstrates significant

advantages in addressing the intricate challenges of head-to-tail imbalance and multi-object recognition.

## 5 CONCLUSION

To tackle the challenges of head-to-tail imbalance and multi-object recognition in long-tailed multi-label image classification, we propose a novel and effective approach, termed Category-Prompt Refined Feature Learning (CPRFL). CPRFL capitalizes on CLIP's text encoder to extract category semantics, leveraging its robust semantic representation capability. This allows for the establishment of semantic correlations between the head and tail classes. The derived category semantics are then utilized as category-prompts, facilitating the decoupling of category-specific visual representations. Through a series of dual-path gradient back-propagations, we refine these prompts to effectively mitigate the visual-semantic domain bias. Simultaneously, the refinement process aids in purifying the category-specific visual representations under the guidance of the refined prompts. To our knowledge, this is the pioneering work to leverage category semantic correlations for mitigating head-to-tail imbalance in LTMLC, offering an innovative solution tailored to the unique characteristics of the data.

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
