# OpenReview forum: "Category-Prompt Refined Feature Learning for Long-Tailed Multi-Label Image Classification"
_acmmm.org/ACMMM/2024/Conference — MM2024 Poster_

### Official Review · Reviewer_j2yu · 2024-05-19

**Rating:** 4
**Confidence:** 2

**Summary:**

This paper proposes a very simple method that uses the text-encoder of CLIP to extract initial category semantics, and an interaction module between categories and visual information to update the category semantics considering both object relationships and visual inputs.
Then the input category semantics of the interaction module are used as classifiers to tag the output updated category semantics.

**Strengths:**

1. A simple method to learn relationship information from CLIP to classify tail categories of LTMLC.
2. Not only the CLIP, but the GloVe embedding also works for the proposed method with only a little performance drop.

**Limitations:**

1.  The RW is a useful trick to improve performance, which is not the concern of this paper. Despiting the RW, the results of the penultimate line in Table 3,  are inferior on tail categories compared with the results in Table 1.
2. Though the paper only uses the text encoder of CLIP discarding the image encoder,  the text encoder has been trained along with the image encoder using text image paired data. So claims in lines 204-207 are somewhat untrustworthy.
3. eq.(7) missing the labels Y.

**Suitability:**

2

---

### Official Review · Reviewer_yD4b · 2024-05-20

**Rating:** 3
**Confidence:** 3

**Summary:**

The paper proposes a novel method, Category-Prompt Refined Feature Learning (CPRFL), to address the challenges of Long-Tailed Multi-Label image Classification (LTMLC). The primary goals are to mitigate the imbalanced data distribution and enhance multi-object recognition. CPRFL utilizes semantic correlations between different categories and decouples category-specific visual representations for each category. This is achieved by initializing category-prompts from pretrained CLIP embeddings and refining them through a progressive Dual-Path Back-Propagation mechanism. The approach aims to suppress negative samples across all classes using Asymmetric Loss, thereby improving head-to-tail recognition performance. The effectiveness of CPRFL is validated through extensive experiments on two LTMLC benchmarks, demonstrating its superiority over existing methods.

**Strengths:**

* __Adequate Evaluation:__ The paper provides comprehensive experimental validation of the proposed method. CPRFL is tested on two well-known LTMLC benchmarks, COCO-LT and VOC-LT, and the results demonstrate significant improvements over state-of-the-art methods. The experiments are thorough, with performance metrics (mean Average Precision) reported for overall, head, medium, and tail classes, highlighting the method's effectiveness across different categories. Additionally, ablation studies and comparisons with various baseline methods further substantiate the robustness and superiority of CPRFL.
* __Clarity:__ The paper is generally well-written, with clear explanations of the methodology, experimental setup, and results. The use of figures and diagrams, such as the overall framework of CPRFL and the visual-semantic interaction process, aids in understanding the complex interactions and processes involved. The structured presentation of contributions and detailed description of the dual-path back-propagation mechanism enhance the readability and comprehensibility of the paper.

**Limitations:**

* __Lack of Detailed Comparison with Similar Methods:__ The paper does not thoroughly compare CPRFL with other CLIP-based methods (such as https://arxiv.org/pdf/2305.04536). While it mentions avoiding direct comparisons due to the potential pretraining bias of CLIP's image encoder, it is essential to include a detailed discussion on how CPRFL differs from or improves upon these methods. This omission leaves a gap in understanding the relative advantages of CPRFL over similar state-of-the-art techniques.
﻿* __Handling of Negative-Positive Sample Imbalance:__ Although the paper adopts Asymmetric Loss to address the negative-positive sample imbalance, there is limited discussion on why this particular loss function was chosen over others and how it compares to other imbalance-handling techniques in detail. Additionally, the paper does not explore the impact of different settings of the Asymmetric Loss parameters (γ+ and γ−) on the final performance, which could be crucial for fine-tuning in various applications.

**Suitability:**

2

---

### Official Review · Reviewer_uyGW · 2024-05-20

**Rating:** 2
**Confidence:** 3

**Summary:**

This paper introduces the  Category-Prompt Refined Feature Learning (CPRFL) to address Long-Tailed Multi-Label image Classification (LTMLC).The authors introduced the semantic prior and conducted tests on two datasets with varying configurations. Furthermore, the authors illustrate that the proposed approach outperforms others.

**Strengths:**

1. This paper is generally well-structured and articulated.
2. The authors have conducted extensive experiments across various datasets and settings, achieving strong performance.

**Limitations:**

1. From Table 3, the effect of PI network improvement is limited, and the authors mention CoOp[1],CoCoOp[2] in the supplemental materials, a more interesting ablation study is to replace the PI with CoOp and CoCoOp while keeping other modules unchanged to show the effectiveness of PI module.

2.  The pretrained text encoder may contain implicitly image information, since CLIP aligns the text embedding space and image feature space, so whether leveraging pretrained text encoder is a too-strong prior knowledge also needs to be validated, and the authors need to ablate on the initialization parameters parameters of text encoders.

3. The author claims the enhanced tail-class performance attributes to the pretrained image encoder, how about the performance finetuning using the pretrained image encoder.

Overall, the the innovation points are unattractive. Although relatively good performance has been achieved, there is a lack of experiments that sufficiently illustrate the effectiveness of the contributions.

[1]  Learning to prompt for vision-language models.
[2] Conditional prompt learning for vision-language models

**Suitability:**

3

---

### Official Review · Reviewer_4Jos · 2024-05-22

**Rating:** 5
**Confidence:** 3

**Summary:**

Authors propose a new method called Category-Prompt Refined Feature Learning (CPRFL) for long-tailed multi-label image classification. A text encoder is firstly used to extract category semantics and further establish the semantic association between the head and tail classes. Then, a progressive dual-path back-propagation mechanism is proposed to iteratively accumulates context-related visual information into the category-prompts, and purify the category-specific visual representations under the guidance of the refined prompts. Lastly, the commonly utilized Re-Weighting strategy is incorporated to address the negative-positive imbalance problem in multiple categories.

**Strengths:**

The novelty of this paper primarily reflect in two main aspects: (a) The category semantics are extracted to establish the semantic association between the head and tail classes, and (b) the progressive dual-path back-propagation mechanism iteratively purify the category-specific visual representations and refine the category-prompts.
Comprehensive experiments are conducted to prove the performance of the proposed method, including the quantitative comparison with other methods, the qualitative analysis of retrieval results, and the ablation study. The experimental results show that the proposed method outperforms the compared methods.
The paper is well wrote.

**Limitations:**

(1) The used encoders in the paper should be explained.
(2) Figure 1 in the paper describes the overall framework, which focuses on the long-tail problems. However, how to deal with the multi-label image classification problem is not shown clearly.
(3) In the experimental part, it is better to add some comparative experiments to verify the effectiveness of the proposed method. For example, authors can try other text encoders such as Word2Vec, GloVe, and so on, to prove the superiority of CLIP.
(4) In the conclusion part of this paper, the possible research direction in the future should be suggested.

**Suitability:**

3

---

### Meta-Review · Area_Chair_TNXW · 2024-07-04

**Recommendation:** Accept (Poster)
**Confidence:** 4

**Metareview:**

The paper proposes a Category Prompt Refined Feature Learning (CPRFL) method for Long-Tailed Multi-Label Image Classification (LTMLC). CLIP’s text encoder is used to establis semantic correlations between the head and tail classes, and a dual-path gradient back-propagations is used to refine the prompts.  Three reviewers give positive scores, and two eviews raise their scores after rebuttal. The AC is inclined  to accept.

Quality: Using head-to-tail correlation to mitigate the imbalance of long-tail problems is reasonable, and the proposed method is effective.

Clarity: The paper is well written but it should  be re-organized to fill the blank of lines 149-152.

Originality: The motivation of the paper is good, and the method is simple yet effective.

Significance： The proposed method is simple and effective, and thus can be applied to other long-tailed tasks and real-world scenarios.